# The Role of Oncogenes and Redox Signaling in the Regulation of PD-L1 in Cancer

**DOI:** 10.3390/cancers13174426

**Published:** 2021-09-02

**Authors:** Christophe Glorieux, Xiaojun Xia, Peng Huang

**Affiliations:** State Key Laboratory of Oncology in South China, Collaborative Innovation Center for Cancer Medicine, Sun Yat-sen University Cancer Center, Guangzhou 510060, China; xiaxj@sysucc.org.cn (X.X.); huangpeng@sysucc.org.cn (P.H.)

**Keywords:** PD-L1, oncogenes, growth factors, ROS, redox homeostasis, cancer, combination therapy

## Abstract

**Simple Summary:**

PD-L1 is an important immune checkpoint molecule that is expressed in cancer tissues at various levels and plays a major role in tumor evasion from immune system. Therefore, understanding the mechanisms by which PD-L1 expression is regulated in tumor environment is important for the design of new therapeutic strategies to overcome tumor immune escape and improve the outcome of cancer treatment. Recent studies suggest that genetic, epigenetic, and transcriptional factors as well as posttranscriptional and posttranslational processes are crucial regulators of PD-L1 expression in tumor cells. This review focuses on two newly described regulations of PD-L1 mediated by oncogenes, and redox homeostasis. In this context, a recent research work describes the important role of oncogenic K-ras, growth factor receptors (FGFR1 and EGFR) and ROS in regulating PD-L1 expression in pancreatic cancer cells and aids significant new mechanistic insights in this important area.

**Abstract:**

Tumor cells can evade the immune system via multiple mechanisms, including the dysregulation of the immune checkpoint signaling. These signaling molecules are important factors that can either stimulate or inhibit tumor immune response. Under normal physiological conditions, the interaction between programmed cell death ligand 1 (PD-L1) and its receptor, programmed cell death 1 (PD-1), negatively regulates T cell function. In cancer cells, high expression of PD-L1 plays a key role in cancer evasion of the immune surveillance and seems to be correlated with clinical response to immunotherapy. As such, it is important to understand various mechanisms by which PD-L1 is regulated. In this review article, we provide an up-to-date review of the different mechanisms that regulate PD-L1 expression in cancer. We will focus on the roles of oncogenic signals (c-Myc, EML4-ALK, K-ras and p53 mutants), growth factor receptors (EGFR and FGFR), and redox signaling in the regulation of PD-L1 expression and discuss their clinical relevance and therapeutic implications. These oncogenic signalings have common and distinct regulatory mechanisms and can also cooperatively control tumor PD-L1 expression. Finally, strategies to target PD-L1 expression in tumor microenvironment including combination therapies will be also discussed.

## 1. Introduction

The discovery of molecules that act as regulators of the immune system opened a new area in tumor therapy. James Allison, recipient of the 2018 Nobel prize, and his colleagues pioneered the concept of immunotherapy for cancer targeting these “immune checkpoints”. In 1996, they demonstrated that the in vivo administration of antibodies against CTLA-4 (cytotoxic T-lymphocyte-associated protein 4) resulted in tumor rejection [1], and was termed the so-called “checkpoint blockade immunotherapies”. Ipilimumab (an anti-CTLA-4 antibody) was the first approved checkpoint blockade immunotherapy by the American Food and Drug Administration (FDA) in March 2011 for the treatment of advanced melanoma. Among important immune checkpoints, PD-1 (programmed cell death-1) is a molecule expressed on T cells that was discovered by Nobel prize winner Tasuku Honjo’s team in 1992 [2]. Later in the 90s, the first ligand of PD-1 receptor was discovered and named PD-L1 (programmed cell death ligand 1), also known as CD274 or B7-H1 [3,4]. A few years later, PD-L2 was reported as a novel PD-1 ligand [5]. The interaction between PD-1 and its ligands induces inhibition of T cells and prevents autoimmunity [4,6]. Based on these findings, PD-L1 antibodies were tested as immune checkpoint blockade. Iwai et al. demonstrated that tumor PD-L1 expression was a novel mechanism of immune escape and blocking the interaction of PD-1/PD-L1 led to tumor regression in several murine tumor models [7]. Given the importance of these molecules for tumor immunity, efforts were made to establish the regulation of their expression and to find better combination therapy in order to kill resistant tumors.

## 2. Structure of Human PD-L1 and Evolution

Since PD-L1 plays an important role in immunity, it was not surprising that no equivalent protein was found in prokaryotes, plants, and primitive eukaryotes. Human PD-L1 is composed of 290 amino acids, shares 40% amino acid identity with human PD-L2 and 70% with its murine ortholog [8]. PD-L1 is a type I transmembrane glycoprotein composed of immunoglobulin-type extracellular domains, a short intramembrane domain, and cytoplasmic tail [9]. All members of the B7 family are predicted to dimerize but it is still uncertain whether PD-L1 could form such homodimers on the cell surface [10,11]. The intracellular domain is very short (length, 30 amino acids) and is highly conserved among species. No biological function and no motif for signal transduction were reported for this protein sequence [12]. The extracellular region is composed of two domains spaced by a short linker sequence. From the cytoplasmic membrane to the *N*-terminus, there is an IgG (immunoglobulin)-like C2 type domain and an IgG-like V type domain [9]. This latter is composed of β strands and mainly interacts with PD-1 receptor and antagonistic antibodies [9,11].

## 3. Tissue Distribution and Subcellular Localization

In various human tissues, the highest PD-L1 mRNA levels were found in the placenta, heart, thymus, and lung. Lower PD-L1 expression was found in the kidneys, spleen, and liver while PD-L1 mRNA was nearly undetectable in the brain, colon, small intestine, testes, and ovary [3,4,13]. In the human digestive system, PD-L1 mRNA levels were higher in tumor tissues than in their respective controls. Conversely, pancreas and colon control tissues had higher PD-L1 mRNA expression [14]. However, the PD-L1 mRNA and protein levels do not always correlate in human and mouse tissues [15,16]. This discrepancy may be explained by the complex regulation of PD-L1 and is further discussed in this review (Section “Regulation of PD-L1 in cancerous tissues”).

PD-L1 is expressed on the cell surface of antigen-presenting cells including dendritic cells and macrophages upon stimulation [17,18,19]. PD-L1 can be also present on activated T and B cells while absent on resting cells [3]. Besides immune cells, PD-L1 can be expressed by fibroblasts [20,21], epithelial [22,23,24], and stromal cells [25]. PD-L1 expression on these cells may also play a critical role in tumorigenesis and participate in immune escape of tumor cells [26,27,28,29]. In addition, PD-L1 was detected in various tumor tissues and tumor cell lines.

Since its receptor is a membrane protein, PD-L1 was first detected on the cell surface of immune cells [3,4]. Through immunoblotting and immunocytochemistry analyses, PD-L1 was localized in the cytoplasm and nucleus of various cancer cells. This aberrant expression of PD-L1 correlated with poor prognosis in colon [30], thyroid [31], and esophageal carcinoma [32]. In breast cancer cells, doxorubicin-induced nuclear PD-L1 translocation and acted as an anti-apoptotic molecule [33]. Moreover, transfection of K562 leukemia cells with a splicing variant lacking IgV domain also showed nuclear and cytoplasmic subcellular localization while membrane localization was observed in cells transfected with wild-type protein [34]. It would be interesting to investigate the proportion of spliced variants and whether anthracyclines might induce splicing of PD-L1 mRNA in cancer cells. However, Polioudaki et al. demonstrated that PD-L1 expression in the cytoplasm or nucleus might be an artifact due to inappropriate sample treatment [35]. Thus, the non-membrane localization of PD-L1 remains to be further investigated.

## 4. Biological Functions and Related Diseases

In order to understand the biological functions of PD-L1, the biological changes in PD-1 expressing cells were investigated because PD-L1 has no intrinsic signaling when engaged with its receptor. Upon its discovery, Dong et al. reported B7-H1 (PD-L1) as a costimulatory molecule but the related mechanisms are still obscure [3]. Freeman and colleagues then demonstrated that the interaction of PD-L1 with its receptor mostly triggered inhibitory signaling in T cells [4]. Indeed, incubation of PD-1^+^ T cells with PD-L1/IgG fusion protein resulted in a decreased proliferation and production of effector cytokine molecules such as interferon-gamma (IFNγ) and interleukin-10 (IL-10) [4]. Studies with PD-1 KO mice further indicated the negative signaling after PD-1/PD-L1 interaction [4]. Similar findings were then observed with T cells from PD-L1 KO mice [36]. Moreover, the proliferation of splenic T cells from PD-1 KO mice was not inhibited when incubated with the PD-L1/IgG fusion protein [4]. Blocking PD-L1/PD-1 interaction in various cell types by using PD-L1 antibodies also supports the negative PD-1 signaling in T cells [7,17,37,38,39,40]. As PD-1 signaling inhibits T cell function, it was not surprising to observe a correlation between PD-1 or its ligands with autoimmune or other immunological diseases. Indeed, the deletion of *PD-1* gene in mice was found to induce lupus-like phenotype [41], cardiomyopathy [42], and type I diabetes [43]. In humans, polymorphism of PD-1 and PD-L1 was found to correlate with the appearance of autoimmune diseases [8]. Several lines of evidence have also demonstrated the PD-1/PD-L1 axis as playing a pivotal role in immunological tolerance, feto-maternal tolerance, transplantation immunity, and against infectious diseases [8]. Finally, PD-L1 expression in tumor and infiltrated immune cells was reported as an immune escape mechanism. Indeed, PD-L1 KO mice bearing MC38 colon tumors achieved tumor regressions [44] and checkpoint blockade showed strong therapeutic activity in multiple corresponding models [7].

## 5. Regulation of PD-L1 in Cancerous Tissues

### 5.1. PD-L1 Expression in Cancers and Role in Tumorigenesis

As previously described, PD-L1 was detected in multiple types of cancers and its expression has been associated with a general mechanism of immune escape [45]. Indeed, PD-L1 expressed by tumor cells can interact with PD-1 receptor and thus lead to T cell inactivation and immunosuppression. Immune escape is now considered as a major hallmark of tumors [46]. The crucial role of PD-1/PD-L1 interaction in the tumor environment and immune escape was demonstrated when the disruption of this interaction by antagonistic antibodies led to tumor regression in many animal models [7] and was further confirmed by successful clinical trials with PD-1/PD-L1 antagonistic antibodies [47]. Consistent with these findings, a positive correlation was found between high PD-L1 tumor expression and poor prognosis in most cancer types [48]. However, the role of PD-L1 in tumor immune response seems rather complex. Although intratumoral PD-L1 expression by immunohistochemistry staining was approved by FDA as a prediction factor for anti-PD-1 therapy response, PD-L1 has been found in some studies to have no significant predictive value in predicting clinical responses to immunotherapies in numerous studies [49,50,51,52]. As such, the prognostic value of PD-L1 expression in predicting clinical response to immune checkpoint blockades remains debatable and predicting responses to immunotherapy is still a major challenge. In the clinical situation, blocking PD-1/PD-L1 interaction exhibited significant therapeutic effects for tumors such as melanomas and lung cancers [53]. However, treatment regimens including PD-1/PD-L1 antibodies as monotherapy or in combination with standard therapies failed to care for patients suffering from pancreatic [54] or colon cancer [55], leading to recurrence [56]. Moreover, chemotherapies may induce elevated PD-L1 expression in tumor cells [57]. The complex role of PD-L1 is further illustrated by its co-regulation with immune responses such as IFNγ and antigen stimulation. PD-L1 may function as a negative regulator of immune response and inflammation occurs in conjunction to the cellular sensing and response to these external cues. In this case, PD-L1 is associated with “hot” tumor microenvironment and immune-responding cancer cells. Moreover, PD-L1 could function as a secondary resistance mechanism in immune-rich tumors. Immune-deprived or “cold” tumors, which are more challenging to treat, usually do not exhibit high PD-L1 expression, as there is no selective advantage for cancer cells in expressing it [58]. Major efforts are currently undertaken to understand the mechanisms of resistance and to find better combination treatment. The resistance to immunotherapies could be explained by several mechanisms. Among them, the absence of antigen stimulation signal, expression of other immune checkpoints, the influence of tumor microenvironment, the initial proportion of tumor-infiltrating lymphocytes (TIL), the presence of immunosuppressive cells, and dynamic tumor PD-L1 expression influence the efficacy of immunotherapies [56].

In this context, understanding the detailed mechanism of PD-L1 regulation could be one of the solutions to overcome resistance. The regulation of PD-L1 is very complex and PD-L1 expression can be controlled by multiple mechanisms (Figure 1). Genetic alterations, epigenetics, transcriptional, posttranscriptional, and posttranslational regulations of PD-L1 have been extensively studied and reviewed, and thus, we will only briefly touch on these regulatory mechanisms in the next sections but will focus mostly on their most recent findings; particularly the three newly described regulations of PD-L1 expression mediated by oncogenes, growth factors, and redox homeostasis.

### 5.2. Genetic Alterations

Human *CD274* gene (PD-L1) is located on the short arm of chromosome 9 (9p24.1). Seven exons encode the transmembrane, cytoplasmic, and extracellular domains. The promoter has CpG methylation sites along the 5′-untranslated region (5′-UTR) and exon 1, while translation starts from exon 2 [59]. Moreover, different gene variants were recently reported [59]. Gene amplification, loss of heterozygosity, and deletion of chromosomal arms are common genetic alterations detected in tumor cells [60]. Since PD-L1 is often upregulated in tumor tissues, particular attention was given to observe a possible link between its expression and *CD274* gene amplification. Amplification of chromosome 9p24.1 in Hodgkin lymphoma and B cell lymphoma was linked to PD-L1 upregulation and its neighbor gene *JAK2* (Janus kinase 2), which also induced PD-L1 transcriptional activation [61]. In Hodgkin lymphoma, 36% had *PD-L1/PD-L2* gene amplification and 56% were characterized by a copy gain [62]. High level of 9p24 amplicon was found in 30% of triple-negative breast cancer (TNBC), 5% of glioblastomas, and 3% of colon carcinomas [63]. However, no association was further observed between PD-L1 expression and copy-number gain status in TNBC [64]. The chromosomal region 9p24 includes the loci for PD-L1, PD-L2 and JAK2. Hence, these molecules are overexpressed in cells harboring a genomic gain and were associated with poor prognosis. Whereas most non-small cell lung cancer (NSCLC) cells lack PD-L1 expression, a subset population (2%) was characterized by a genomic gain and was particularly sensitive to immune checkpoint blockade [65]. Moreover, genomic alterations also correlated with PD-L1 expression in NSCLC [66]. Indeed, the copy gain was found to be correlated with high PD-L1 expression, whereas loss of chromosomal arm 9p24 was associated with low PD-L1 expression [65]. Most *CD274* gene amplifications are caused by genomic rearrangements and do not affect the open reading frame, thus, leading to high-level expression of PD-L1. Conversely, PD-L1 expression did not always correlate with gene copy number in certain cancers, suggesting that transactivators might be lacking in these tumors. Therefore, immunohistochemistry PD-L1 analyses would better predict response to immunotherapies than comparative genomic hybridization or fluorescence in situ hybridization analyses.

Many studies reported a significant effect of PD-L1 polymorphism on patients’ outcomes. Studies from Kula et al. on PD-L1 polymorphism showed their influence on cancer stage, effectiveness of chemotherapy, and prognosis after tumor resection [67]. Most of these polymorphisms were located in the promoter region, 3′-UTR (3′-untranslated region), and introns. They might alter the binding of transcription factors (i.e., Specific Protein 1) or micro-RNAs, and could influence the expression of PD-L1.

### 5.3. Epigenetic Alterations

Epigenetic modifications can contribute to or prevent carcinogenesis by altering the gene expression profile. In this context, epigenetic changes have been associated with profound modification of PD-L1 expression in cancer cells [59,68,69]. DNA methylation and histone acetylation/methylation are the main epigenetic modifications. Five CpG islands were found in the PD-L1 promoter [59] and the quantification of methylated *PD-L1* gene was proposed as a prognostic tool [70]. First, the hypomethylation of the PD-L1 promoter was detected in melanoma [71] and this phenomenon was correlated with an interferon signaling phenotype in this type of tumor [72]. The proportion of hypomethylated PD-L1 promoter was increased in primary breast cancer and colorectal cancer [73]. In contrast, Sasidharan Nair et al. did not observe changes in DNA demethylation between normal colon and cancer tissues [74]. Most of these publications provide evidence showing that a combination with epigenetic therapy and immunotherapy could be a promising approach in cancer treatment; considering that prior research targeting DNA methylation and histone methylation have shown improvements in patients’ outcomes [75]. Moreover, a strategy to increase the methylation of PD-L1, by using fusion protein DNA methyltransferase (DNMT) and Zinc finger domain that binds PD-L1 promoter, decreased PD-L1 function in prostate cancer cells [76]. Although DNMT inhibitors tend to increase PD-L1 levels in most studies, decitabine and quisinostat decreased PD-L1 expression in multiple myeloma [77].

In addition to the change in DNA methylation pattern, PD-L1 expression is tightly regulated by histone methylation or acetylation, modifying chromatin remodeling and therefore gene expression. Indeed, the EZH2 (enhancer of zeste homolog 2), LSD1 (lysine-specific demethylase 1), BET (bromo- and extra-terminal domain) family members, and HDAC (histone acetylases) were found involved in this regulatory process. EZH2 is a subunit of the polycomb repressive complex 2 and a histone methyltransferase. This enzyme mediates gene repression by increasing tri-methylated H3K27 levels around target genes. EZH2 activity correlated with PD-L1 downregulation in hepatocarcinoma [78] and was dependent on the HIF-1α (hypoxia-inducible factor 1 alpha) activation in lung cancer [79]. BET family members are a class of epigenetic readers that bind acetylated histones and facilitate the recruitment of co-activators complexes. The BET inhibitor JQ1 repressed *PD-L1* gene transcription in prostate cancer and potentiated with anti-CTLA-4 [80]. Moreover, JQ1 was associated with decreased PD-L1 expression in nasopharyngeal cancer [81] and oral squamous cell carcinoma [82]. Another BET inhibitor, PLXS1107, showed similar effects and potentiated immunotherapy in melanoma [83]. In pancreatic stellate cells, BET inhibition was directly linked to the IFNγ-mediated PD-L1 expression [84]. HDAC3 silencing led to a decreased DNMT1 expression, PD-L1 upregulation, and potentiated PD-L1 antibody [85]. HDAC inhibitors showed similar effects and synergized with PD-1/PD-L1 checkpoint blockade [86]. Conversely, HDAC6 was found to be positively correlated with PD-L1 expression, suggesting a histone-independent mechanism [87]. Nevertheless, HDAC6 inhibition sensitized ovarian tumors to anti-PD-L1 therapy [88]. LSD1 has histone demethylase activity and acts as a transcriptional activator. In triple-negative breast cancer, LSD1 was inversely correlated with PD-L1 expression [89].

### 5.4. Signaling Pathways and Transcriptional Regulation

The involvement of signaling pathway activation and the role of transcription factors were investigated in regard to PD-L1 regulation in cancer cells. The first signaling pathway to be reported as a PD-L1 regulator was Akt/PKB (protein kinase B). This study suggested that a loss of PTEN (phosphatase and tensin homolog) subsequently activated PI3K (phosphoinositide 3-kinase)/Akt signaling and PD-L1 upregulation in glioma [90]. These findings were further confirmed in other models [91,92] and the Akt-mediated mechanism was mostly related to PD-L1 protein stability rather than transcriptional regulation. Akt is mainly activated through receptor tyrosine kinases such as EGFR (epidermal growth factor receptor) and FGFR1 (fibroblast growth factor receptor 1) and are further discussed below.

Activated by different sources of stimuli, the MAPK (mitogen-activated protein kinase) [93,94], AMPK (adenosine 5‘-monophosphate (AMP)-activated protein kinase) [95], JAK/STAT (signal transducer and activator of transcription) [96,97], GSK3 (glycogen synthase kinase-3)/β-catenin/ZEB1 (zinc finger E-box binding homeobox 1) [98], and NF-κB (nuclear factor kappa-B) [99,100] pathways are also able to control *PD-L1* gene expression. They may crosstalk to control PD-L1 expression [101]. In addition, several transcription factors are known to bind to PD-L1 promoter and regulate its transcriptional activity. Indeed, Myc (Myc proto-oncogene) [102], STAT3 [103], NF-κB [99], YY1 (Yin Yang 1) [104], and AP-1 (activator protein 1) [105,106] were found to bind the promoter and positively regulate PD-L1 expression in cancer cells. Moreover, tumor microenvironment conditions such as hypoxia were found to enhance PD-L1 expression [107] and HIF-1α could directly bind to the promoter to control its expression [108]. Recently, a super-enhancer region was identified in PD-L1 and PD-L2 promoters and could be potentially responsible for tumor immune escape [109].

### 5.5. Posttranscriptional and Posttranslational Regulations

Although these changes in PD-L1 mRNA levels can be imputed to gene transcription regulation, many studies have demonstrated that PD-L1 could be regulated at the post-transcriptional level. After treatment with actinomycin D, the half-life of PD-L1 mRNA was about 3 h in MDA-MB-231 [110] and HeLa cells [111]. In these two studies, TTP (tristetraprolin) and AUF1 (AU-rich element-binding protein 1) bounded to 3′-UTR and modulated mRNA stability. 5′-UTR and 3′-UTR of PD-L1 mRNA possess many regulatory elements. A plethora of micro-RNAs can, directly or indirectly, alter PD-L1 expression in cancer and immune cells by interacting with these regulatory sequences [112].

The *N*-glycosylation is the most important posttranslational modification (PTM) for PD-L1 and can lead to protein stabilization and immunosuppression [113]. Indeed, the half-life of glycosylated PD-L1 was about 12 h whereas the non-glycosylated form was about 4 h. Targeting PD-L1 glycosylation with monoclonal antibodies [114] or the glycosyltransferase GLT1D1 (glycosyltransferase 1 domain containing 1) [115] has been found to enhance tumor immune response in TNBC and B cell lymphoma, respectively. Other modifications such as phosphorylation, ubiquitination, and palmitoylation were also reported [69,116] but the characterization of these PTM still remains to be elucidated. In this context, CMTM4/6 (chemokine-like factor-like MARVEL transmembrane domain-containing family member 4/6) protects PD-L1 from ubiquitination and degradation in various cancer cell lines and dendritic cells [117]. In addition, Zhang et al. demonstrated that cyclin D and CDK4 (cyclin-dependent kinase 4) could destabilize PD-L1 protein via cullin-3/SPOP (Speckle type BTB/POZ protein)-dependent mechanism [118].

Akt pathway is one of the primary signaling pathways to regulate PD-L1 expression and was first documented to upregulate PD-L1 expression through translation regulation [90]. Many reports support such regulation, but the underlying regulatory mechanisms are still unknown. Recently, estrogen was found to stabilize PD-L1 mRNA in cancer cells via an Akt-dependent activation [119].

Regarding PD-L1 protein stability, metformin, in combination with immunotherapy, demonstrated promising therapeutic activity by inhibiting PD-L1 through an AMPK-dependent mechanism and ER (endoplasmic reticulum)-associated degradation [95,120].

### 5.6. Cytokines

Since immune cells are mainly stimulated or inhibited by secreted factors or by cell-cell contact, many studies investigated the role of such extracellular proteins in order to regulate PD-L1 expression. Freeman et al. demonstrated that IFNγ could upregulate PD-L1 in monocytes and dendritic cells [4]. A few years later, it was reported that IFNγ also stimulated PD-L1 expression in hepatoma cancer cells [34]. It was suggested that high production of IFNγ (secreted by cytolytic CD8^+^ T cells, Natural killer, and effector CD4^+^ T cells) had antitumor effects while low IFNγ levels upregulated PD-L1 and triggered immune evasion [34]. Many reports further demonstrated the role of IFNγ in upregulating PD-L1 in cancer cells and other cell types [100,121,122], mainly through JAK/STAT activation [96].

Other cytokines are able to stimulate PD-L1 expression. Indeed, IFNγ and TNFα (tumor necrosis factor-alpha) can together activate NF-κB and PD-L1 expression in blasts of myelodysplastic syndromes [100]. Type I IFN can also positively control PD-L1 expression in myeloid-derived suppressor cells [123] and transforming growth factor-beta (TGF-β) induced PD-L1 expression in dendritic cells [124]. Moreover, interleukin-6 (IL-6) [125] and granulocyte-macrophage colony-stimulating factor (GM-CSF) [126] are also able to upregulate PD-L1 in stroma cells.

While the role of cytokines is well established, growing evidence showed growth factors and their receptors also played an important role in immune escape and tumor progression. In the following sections, we are to focus on the roles played by EGFR and FGFR receptors.

### 5.7. Oncogenes

The presence of activated oncogenes seems to reduce the clinical response to immune checkpoint inhibitors in some studies [127]. In this context, studies that examined a putative link between PD-L1 expression, oncogenic proteins and patient survival led to confusing and discordant conclusions. For example, bioinformatics analyses mainly compare mRNA levels among different tissues and cancer types. Given that PD-L1 expression may be regulated at the translational level, findings obtained by such databases are therefore difficult to interpret. In this context, immunohistochemistry analyses may be more reliable, but such analyses remain semi-quantitative and artifacts or many unknown factors could explain a change in PD-L1 protein levels. We thereby discuss publications that showed strong evidence that oncogenes control PD-L1 expression in cancer cells and the related regulatory mechanisms.

#### 5.7.1. Fibroblast Growth Factor Receptors

In regard to the FGFR-mediated PD-L1 regulation (Figure 2), McNiel and Tsichlis first identified a positive correlation between FGF2, FGFR1, Akt3 and immune checkpoints (CTLA-4, PD-1 and PD-L1) in bladder carcinoma [128]. Later, a correlation between FGFR2 and PD-L1 was found in colorectal cancer [129]. FGF7 bound to FGFR2 in SW480 cells and induced *PD-L1* gene transcription through JAK-STAT activation. Consistent with these findings, lung tumors expressing mutant FGFR2 or K-ras (Kirsten ras GTPase) exhibited an increase in PD-L1 levels [130]. In this study, the pan-FGFR inhibitor erdafitinib decreased PD-L1 protein levels, increased CD3^+^ T cell infiltration, and synergized with PD-1 antibody. Moreover, ODM-203, a potent inhibitor of FGFR and VEGFR (vascular endothelial growth factor receptor), was associated with an increased number of intratumoral CD8^+^ T cells and NK cells [131]. In line with these results, VEGF and basic FGF inhibited the secretion of IFNγ (interferon gamma) and granzyme B by T cells [132]. Lenvatinib, another VEGFR and FGFR inhibitor, was able to reverse the immunosuppressive effect mediated by growth factors and synergized with anti-PD-1 antibody in a colon cancer model. Finally, FGF1/2 bound to FGFR1 and induced PD-L1 mRNA and protein levels in pancreatic cancer cells [133]. A positive correlation was also found between FGFR1 and PD-L1 in pancreatic cancer tissues. FGFR1 activation stimulated Akt/PKB pathway and led to protein stabilization. For now, mechanisms that explained FGFR1-mediated *PD-L1* gene transcription are unknown and most of the known regulators (i.e., STAT3) were excluded. The FGFR1 inhibitor PD173074 also decreased PD-L1 protein levels, but not mRNA levels, in in vitro and in vivo mouse pancreatic tumor models. Human and mouse FGFR1 KO cell lines exhibited lower PD-L1 expression compared to wild-type cells. Moreover, mouse pancreatic FGFR1 KO tumors grew slower due to PD-L1 downregulation and an increase in intratumoral CD8^+^ T cells. Consistent with these findings, FGFR1 KO breast cancer-bearing mice exhibited a higher proportion of intratumoral CD8^+^ T cells, a lower number of myeloid-derived suppressor cells (MSDC), and became more sensitive to PD-L1 antibody [134]. Moreover, the covalent FGFR inhibitor FIIN4 demonstrated similar effects with FGFR1 KO cells. In concordance with these findings, 40% of colorectal cancers have mutant K-ras and therefore oncogenic signals might alter FGFR1 and FGFR2 activities to enhance immunosuppression.

In clinical trials, immune checkpoint blockades have limited efficacy in luminal subtypes of urothelial cancer with FGFR3 mutations. Only a small proportion of patients with FGFR alterations responded to initial PD-1/PD-L1 inhibitors [135]. It was also demonstrated that FGFR3 mutation status was not a biomarker of resistance to immunotherapy in metastatic urothelial cancer [136]. Since the FGFR-dependent regulation of PD-L1 expression was discovered only recently, there has been no report on a putative link between FGFR1/2 mutation status or overexpression and response to PD-1/PD-L1 inhibitors. Currently, there is no data showing a direct correlation between mutant FGFRs and PD-L1 expression. It would be interesting to investigate the possibility that such mutants might affect the expression of PD-L1 in cancer cells. Further studies are therefore needed to elucidate the role of FGFRs in modulating tumor immune response. Interestingly, combination of FGFR inhibitors and immunotherapy showed promising therapeutic effect in multiple solid tumors in clinical trials [137,138,139], and the underlying mechanisms of action still need to be fully understood.

#### 5.7.2. Epidermal Growth Factor Receptor

Figure 3 shows the role of other oncogenes to regulate PD-L1 expression in cancer cells including c-Myc, p53, EML4-ALK, K-ras and EGFR.

The first evidence that demonstrated the crucial role of growth factors to control PD-L1 expression was revealed in mutant EGFR NSCLC [140]. In that study, the EGFR inhibitor, gefitinib, was able to decrease PD-L1 cell surface expression. Most of the results were obtained in lung cancer models because EGFR is often mutated in this cancer type. By using exogenous EGF and pharmacological inhibition of EGFR, it was shown that EGFR could control PD-L1 expression through a MAPK/Akt/STAT-dependent mechanism in NSCLC [141,142,143,144,145]. MAPK also increased PD-L1 mRNA levels [143]. The E3 ubiquitin ligases Cbl-b (Casitas B-lineage lymphoma proto-oncogene-b) and c-Cbl (Cbl proto-oncogene C) inhibited PD-L1 by inactivating STAT, AKT, and ERK signaling pathways [144]. Moreover, the EGFR inhibitor osimertinib suppressed PD-L1 expression at the posttranslational level via a GSK3-dependent mechanism [146].

In addition to observations in lung cancer, EGFR has also been found to activate the MAPK/Akt/STAT/Myc signaling pathways to upregulate PD-L1 expression in head and neck [147], esophageal [148,149], gastric [150], salivary adenoid cystic carcinoma [151] and EGFR-positive cancers [152]. IGFBP2 (insulin-like growth factor binding protein 2) activated STAT3 phosphorylation by co-localizing with cytoplasmic EGFR. This association facilitated EGFR nuclear accumulation and activated EGFR phosphorylation. The recruitment of STAT3 to the PD-L1 promoter induced its transcriptional activation in melanoma [153]. EGFR activation enhanced CSN6 (COP9 signalosome 6) expression in a MAPK-dependent manner in glioblastoma. CSN6 inhibited PD-L1 protein degradation [154]. Moreover, glutamine deprivation resulted in the activation of EGFR signaling through MAPK and c-Jun to regulate PD-L1 expression in renal cancer cells [155].

Finally, the expressions of EGFR and EGF were increased when mutant K-ras was induced in the 293T-K-ras-inducible system. Incubation with EGF alone was sufficient to increase PD-L1 expression in 293T and pancreatic cancer cell lines. Pharmacological inhibition of EGFR with gefitinib and afatinib abolished the K-ras-mediated PD-L1 upregulation. However, afatinib did not change PD-L1 expression in pancreatic cancer cell lines, suggesting that EGFR is not the main factor responsible for immunosuppression in pancreatic cancer [133].

Akbay et al. first demonstrated that PD-L1 expression was upregulated in mutant EGFR-driven lung cancer and might be also associated with K-ras mutation [140]. As previously described, wild-type EGFR has the ability to control PD-L1 expression. However, other studies reported that mutant EGFR was also involved in the regulation of the immune checkpoint. Indeed, gefitinib and erlotinib inhibitors were found to abolish PD-L1 cell surface expression in mutant EGFR NSCLC [156]. Moreover, gefitinib suppressed NF-κB activation and decreased PD-L1 cell surface expression [157]. Finally, ERBB2/3 mutants activated Akt signaling and augmented PD-L1 expression in gallbladder carcinoma [158].

Finally, NSCLC patients with EGFR mutations show poor response to anti-PD-1/PD-L1 treatment [159], and the mechanisms responsible for the lack of response to immunotherapy in such patients still remain unclear and need to be investigated. Moreover, combination of tyrosine kinases inhibitors and immunotherapy did not show synergistic effects in NSCLC patients in recent clinical trials [160,161]. Further efforts are therefore needed to evaluate different types of drug combinations, administration sequences, and side effects to identify more effective immunochemotherapy for this subset of patients.

#### 5.7.3. c-Myc

Depending on the cellular context, the c-Myc oncogene could be a positive or negative regulator of PD-L1 expression. Durand-Panteix et al. were the first to demonstrate that c-Myc was a strong repressor of PD-L1 in EBV (Epstein–Barr virus)-immortalized B cell via STAT1 inhibition. In these cells, c-Myc mainly blocked PD-L1 surface membrane export by decreasing actin polymerization [162]. In addition, Myc inhibition led to STAT1 activation and an increase in PD-L1 expression induced by IFNγ in hepatocarcinoma [163]. When co-cultured with T cells, an inverse correlation between c-Myc and PD-L1 was found in diffuse large B cell lymphoma (DLBCL) [164]. Finally, Myc inhibitors increased the expression of the immune checkpoint on tumors and sensitized them to PD-1 antibody [165], and these new findings need further evaluation of such drug combination in clinically relevant settings. Afterward, it was shown that Myc transactivation activity could directly bind to *PD-L1* and *CD47* promoters [102], and a positive correlation between c-Myc and PD-L1 in B cells was observed [166].

In serous papillary endometrial cancer, STAT1 upregulated c-Myc and PD-L1 protein levels [167]. In addition, STAT3 and c-Myc cooperated to induce PD-L1 in DLBCL [168]. Mesenchymal stem cells secreted IL-8 which in turn activated STAT3 and mTOR (mammalian target of rapamycin), to induce c-Myc and PD-L1 expressions in gastric cancer cells [169]. Bridging integrator-1 (BIN1), a c-Myc adaptor protein, inhibited Myc function and EGFR signaling in NSCLC, therefore suppressing PD-L1 expression [170]. Moreover, c-Myc induced *PD-L1* gene transcription in neuroblastoma [171], lung [172], gastric, pancreatic [173], and esophageal [174] cancers. Moreover, Myc synergized with K-ras to induce tumor immunosuppression [175,176]. In these studies, Myc was shown to control PD-L1 at transcriptional and translational levels. In this context, a positive correlation was found between Myc and PD-L1 protein but not with mRNA in DLBCL [177]. These data are very interesting because they demonstrated that Myc could directly bind to the promoter and was predicted to have transactivation activity. The Myc-mediated PD-L1 translational regulation still remains to be fully elucidated. Altogether, it is still unclear how Myc might repress or activate *PD-L1* gene transcription. The repressor Myc activity on PD-L1 might be an indirect effect or Myc might cooperate with other transcription factors and bring corepressor complexes.

Currently, limited information is available regarding the correlation between Myc overexpression and response to PD-1/PD-L1 inhibitors in tumors such as B cell non-Hodgkin lymphomas. Further studies are therefore needed to evaluate the possibility of administrating immunotherapies in patients with Myc overexpression.

#### 5.7.4. Protein p53

The protein p53 (protein of 53 kDa) is usually considered as a tumor suppressor gene but many p53 mutants act as oncogenic proteins [178]. Similar to Myc-mediated regulation, experimental findings suggest a double-faced role of p53. First, the protein p53 was shown to participate in PD-L1 regulation and was first considered as the negative regulator. Indeed, a negative correlation between PD-L1 and p53 was previously shown in hepatocarcinoma [179]. Mechanistically, higher expression of PD-L1 was observed in p53 knockout colon and lung cancer cells. The oncogenic protein regulated the expression of microRNA miR34 that bonded and led to the destruction of PD-L1 mRNA [180]. In line with these findings, wild-type p53 promoted cytotoxic T lymphocytes induced cancer cell death via miR34a and PD-L1 inhibition [181].

However, p53 is also viewed as a positive regulator of PD-L1 expression. Indeed, mutant K-ras and p53 activated ARF6 (ADP Ribosylation Factor 6)-ASAP1 (ArfGAP with SH3 domain, ankyrin repeat, and PH domain 1) axis through PDGFR (platelet-derived growth factor receptor) signaling to control PD-L1 expression in pancreatic cancer cells [182]. The mutant p53 protein was at the origin of the elevated OXSR1 (oxidative stress-responsive kinase 1) expression that positively correlated with PD-L1 and the number of TIL [183]. Moreover, p53 inhibition led to a diminution of IFNγ-induced PD-L1 upregulation through Jak2 inhibition in melanoma [184]. Obviously, more efforts are needed to decipher the molecular mechanisms, mediated by p53, in regulating PD-L1 expression in cancer cells.

In the clinical context, patients with p53 mutations showed better response to immunotherapy compared to patients harboring wild-type p53 [185], but not all p53 mutations are equal in predicting therapeutic efficacy in patients treated with PD-1/PD-L1 inhibitors [186]. Further studies are needed to understand the mechanisms of resistance of wild-type p53 tumors to immunotherapies. Interestingly, a recent study demonstrated that MDM2 (murine double minute 2) inhibitors could activate p53, induce IFN response and synergize with immunotherapy [187]. This approach could potentially transform cold tumors into hot tumors and need further consideration for clinical study.

#### 5.7.5. EML4-ALK

Lung cancer is often characterized by the presence of several oncogenes including mutant EGFR and K-ras. Another frequent anomaly in lung cancer is the EML4-ALK (echinoderm microtubule-associated protein-like 4-anaplastic lymphoma kinase) fusion gene. This oncoprotein has shown the ability to enhance PD-L1 expression. Indeed, EML4-ALK stimulated STAT3 binding activity to elevate PD-L1 expression in pulmonary adenocarcinoma. The expression of the immune checkpoint was further enhanced by HIF-1α binding under hypoxic conditions [188]. Moreover, oncogenic signaling activated MAPK and Akt pathways in order to trigger PD-L1 expression in NSCLC [189]. In these two latter studies, silencing ALK suppressed PD-L1 protein levels. In addition, overexpression of EML4-ALK enhanced PD-L1 expression in NSCLC. Inhibition of ALK, which suppressed MAPK and Akt activation, also led to IFNγ production in coculture system with dendritic and T cells [190].

Currently little is known regarding the exact role of EML4-ALK on tumor immune response. A recent retrospective study revealed that approximately 10% of ALK-positive patients responded to immunotherapies [191]. Combination of ALK inhibitors and immunotherapy seems to show efficacy but some patients developed severe hepatotoxicity, and long-term clinical outcomes have not yet been reported [192,193]. Therefore, further studies are needed to identify effective immunotherapy and its combination with chemotherapy to achieve favorable clinical outcome for patients with *ALK* gene rearrangements.

#### 5.7.6. K-Ras

The first evidence that mutant K-ras may regulate PD-L1 was found in NSCLC. In that study, mutant K-ras triggered the activation of MAPK/AP-1/STAT3 signaling in order to induce PD-L1 expression [194]. In line with these findings, mutant K-ras activated MAPK signaling [195] through FRA1 (FOS-related antigen 1) transcriptional activity [196] to augment PD-L1 protein levels in NSCLC. Moreover, higher PD-L1 expression was observed in mutant EGFR and K-ras lung cancer cells through an Akt/mTOR-dependent mechanism [197]. In this same study, PD-L1 expression was higher in mutant K-ras HCT116 colon cancer cells compared to wild-type K-ras HCT116 cells.

In pancreatic cancer cells, K-ras decreased the levels of TGIF1 (TGF-β induced factor homeobox 1) which is a nuclear corepressor of *PD-L1* gene transcription [198]. Moreover, FGFR1 signaling plays an important role in the K-ras-mediated PD-L1 upregulation. Indeed, exogenous FGF or FGFR1 inhibitors were able to modulate PD-L1 expression in pancreatic cancer cell lines. Moreover, there was a correlation between PD-L1 and FGFR1 expressions in mutant K-ras pancreatic tissues and not in wild-type K-ras cancers [133].

The best proof of the involvement of oncogenic K-ras in this regulation was found in K-ras inducible cell systems. By using such in vitro models, oncogenic protein activation was associated with an increase in EGFR and FGFR1 protein levels. Activation of both growth factor receptors stimulated Akt signaling leading to increased PD-L1 mRNA and protein levels [133]. In another in vitro model, K-ras activation led to PD-L1 mRNA stabilization. Mechanistically, K-ras signaling activated the MK2 (MAPK-activated protein kinase 2) enzyme which further inhibited the TTP activity by phosphorylation [199]. Moreover, MAPK enhanced PD-L1 mRNA stability in mutant K-ras NSCLC [143]. Altogether, these data demonstrated a strong immunosuppressive effect mediated by oncogenic signals.

The responses to immunotherapy of NSCLC patients with K-ras mutations are heterogeneous, and it remains difficult to draw solid conclusions because K-ras mutated disease might show different clinical characteristics. Moreover, the co-existence of other genetic events (i.e., mutant p53) may influence response to anti-PD-1/PD-L1 treatment for NSCLC patients [191]. It is worth noting that patients with colorectal cancers with K-ras mutations and pancreatic cancer patients (90% have mutant K-ras) exhibit poor response to immunotherapy [54,55]. The mechanisms of resistance to immunotherapy might not only depend on the presence of specific oncogenes or their mutation status but might also be influenced by the unique tumor microenvironment observed in certain tumor types.

### 5.8. Redox Regulation of PD-L1

Tumor cells frequently generate and secrete reactive oxygen species (ROS) and the resulting oxidative stress in the tumor microenvironment is now emerging as a potent PD-L1 inducer. Many studies have reported high PD-L1 expression associated with high generation of ROS but no direct link between these two observations has been investigated [200,201,202,203,204,205,206,207].

Bailly et al. first described the variable effects of ROS-modulating drugs on PD-L1 [208]. PD-L1 was found to be regulated by NF-κB and HIF-1α, which are transcription factors controlled by redox mechanisms. However, these compounds might have multi-target effects and may explain the differential PD-L1 expression during exposure to such pro-oxidant drugs. Since antioxidants such as *N*-acetyl-cysteine might also have multi-target effects, the use of hydrogen peroxide (H_2_O_2_) could be the best proof-of-concept to study the redox regulation of PD-L1 depicted in Figure 4. In this context, H_2_O_2_ enhanced PD-L1 protein in monocytes [209]. During infection to EBV, TLR7/8 (Toll-like receptor 7/8) signaling elevated PD-L1 through MyD88 (myeloid differentiation primary response 88) and IRAK1/4 (interleukin 1 receptor-associated kinase 1/4) signaling. During viral infection, ROS is generated from TLR/MyD88/IRAK signaling because IRAK inhibition abolished EBV-mediated PD-L1 upregulation. Moreover, induced oxidative stress was associated with p38 (protein of 38 kDa)/JNK (c-Jun *N*-terminal kinase)/NF-κB/STAT3 activation to control PD-L1 expression in infected monocytes. It is still unclear how ROS regulates PD-L1 expression in monocytes, possibly through direct activation, indirect activation (via NF-κB, MAPK), induction of *PD-L1* gene transcription, or translational regulation. Infection with HHV-6B (human herpesvirus 6B) showed similar findings with elevated ROS levels which in turn stimulated STAT1/3 and PD-L1 [210]. In bone marrow-derived macrophages, paclitaxel generated ROS through an unknown mechanism and enhanced the expression of the immune checkpoint. This oxidative stress led to NF-κB activation and *PD-L1* gene transcription [211]. In K-ras-driven cancer cells, H_2_O_2_ or glucose oxidase (an enzyme that generates H_2_O_2_) augmented PD-L1 mRNA and protein levels [133]. ROS induced *FGFR1* gene transcription by a yet to be discovered mechanism. Moreover, ROS could directly activate this FGFR1 receptor. Generation of H_2_O_2_ cannot induce PD-L1 expression in FGFR1 KO cells, suggesting the receptor played a crucial role in this redox regulation. FGFR1 signaling increased PD-L1 protein level through Akt activation. However, the molecular mechanisms of how FGFR1 induced PD-L1 mRNA levels remain to be uncovered. A possible explanation could be that FGFR1 activation might stimulate the binding of transcription factors on the *PD-L1* promoter or might stabilize its mRNA. In agreement with this hypothesis, ROS inhibited TTP enzyme activity in mutant K-ras cancer cells [199]. It would be interesting to investigate whether FGFR1 might modulate TTP enzyme activity and PD-L1 mRNA stability.

## 6. Conclusions

Although PD-L1 inhibitors showed promising therapeutic effects, we raised the question of whether targeting PD-L1 by decreasing or increasing its function in tumor cells is really a good therapeutic option. On one hand, authors have tried to increase PD-L1 expression in cancer cells in order to sensitize cancer cells to PD-1/PD-L1 immune checkpoint blockade with some success in preclinical models. While these studies are important with respect to understanding the mechanisms of PD-L1 regulation and potential clinical implications, artificially inducing PD-L1 expression in cancer may not be a good therapeutic strategy. Indeed, this strategy appears to be a double-edged sword as it could potentially enhance tumor immunosuppression. Tumor tissues with high intrinsic expression of PD-L1 tend to be a reflection of adaptive resistance to initial immune attack and could be overcome by therapeutic antibodies, whereas artificial induction of PD-L1 expression would promote immunosuppression, which should be avoided. On the other hand, decreasing PD-L1 protein (i.e., metformin) levels could be an effective strategy but the use of anti-PD-1/PD-L1 antibodies could lead to similar results and might become obsolete because of the loss of immune checkpoint expression. Therefore, protocol targeting PD-L1 expression could be supplemented with other immunotherapies such as anti-CTLA-4. As explained in this review, the regulation of PD-L1 expression is very complex and controlled via multi-step levels. Selecting one of these steps to alter the immune checkpoint expression might be inefficient because the other regulatory elements might compensate and be hyperactivated. Moreover, PD-L1 surface expression on antigen-presenting cells, stromal cells, and epithelial cells also participate in immune evasion and may be regulated by different mechanisms. However, targeting the protein stability (i.e., Akt inhibitors) or PTM (i.e., monoclonal antibodies) could be a very interesting approach to inhibit the PD-1/PD-L1 interaction between all these cells in the tumor microenvironment.

In the context of clinical relevance, the information regarding the exact role of each individual oncogene on tumor immune response is currently limited and insufficient as a basis to formulate optimal immunotherapeutic strategies. Furthermore, since some tumors may have mutations in multiple genes such as K-ras and p53 in the same cells, this would make it even more complicated to formulate oncogene-specific immunotherapy. Obviously, further studies are needed in this area. However, the upregulation of PD-L1 by ROS/redox signaling and oncogenes such as K-ras is of particular significance, since ROS stress and activation of K-ras are frequently observed in various types of human cancers. Furthermore, because ROS play a major role in upregulation of PD-L1 expression, modulation of redox signaling could be a potential strategy to alter tumor immune microenvironment and improve the efficacy of immunotherapy. Such therapeutic strategy is novel and requires further evaluation in clinically relevant settings.

A combination of anti-PD-1/PD-L1 antibodies or PD-L1 degraders with chemotherapies could be the most feasible and effective manner to overcome resistance and allow a multitarget mechanism of action. However, we should be cautious about the choice of chemotherapies. These drugs should not inhibit immune cell function and have the potential to activate the immune system or induce immune cell infiltration into the tumor environment. In this context, potential immunosuppressive (arsenic trioxide, camptothecin, and fludarabine) and immunostimulant (i.e., melphalan, doxorubicin, paclitaxel, cisplatin) drugs were identified in vitro but the therapeutic effect of such immunochemotherapy regimens including these drugs still need to be investigated in preclinical and clinical studies [212]. Major efforts must be undertaken now to find the best combination therapies and understand in detail the related molecular mechanisms of action.

## Figures and Tables

**Figure 1 cancers-13-04426-f001:**
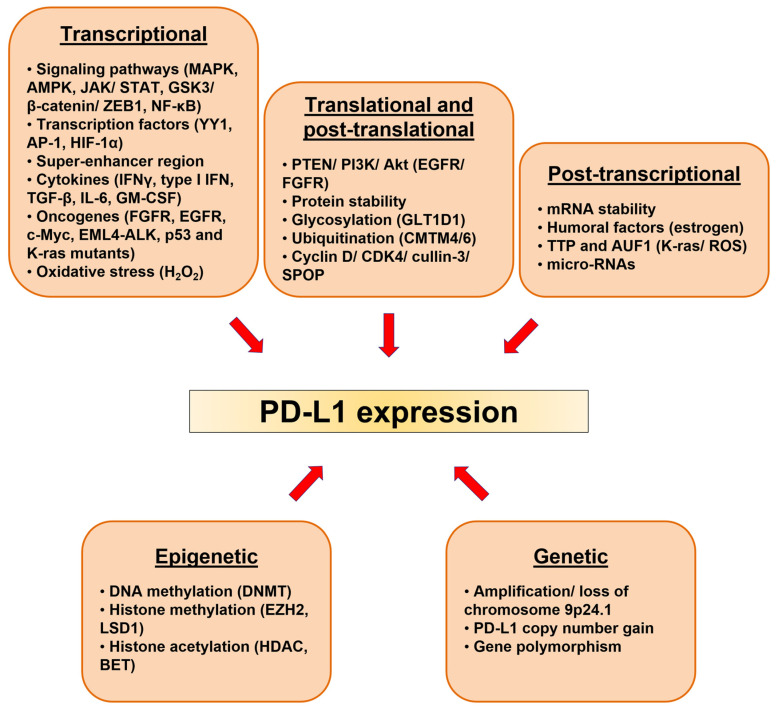
The expression of PD-L1 is regulated at various levels.

**Figure 2 cancers-13-04426-f002:**
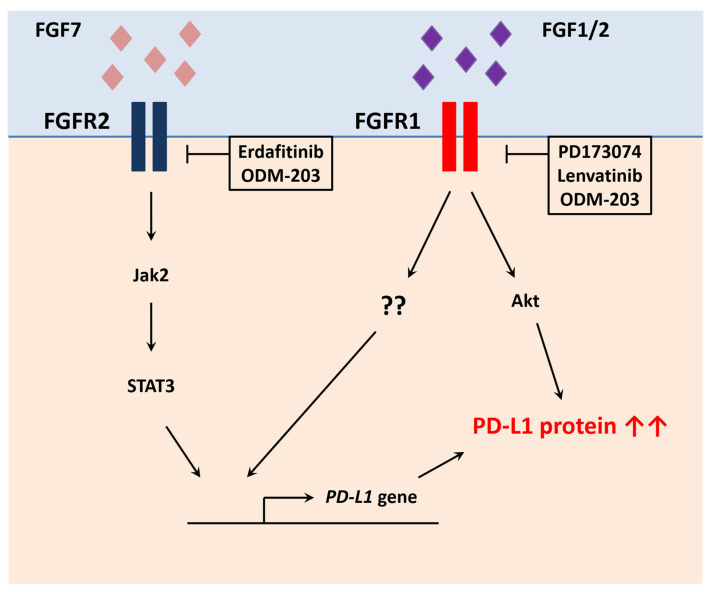
FGFR member family members positively regulate PD-L1 expression in cancer cells. FGFR1 signaling activates Akt signaling and PD-L1 stabilization. FGFR1 also enhances *PD-L1* gene transcription by a yet to be discovered mechanism. FGFR2 activates Jak2/STAT3 signaling, which in turn causes PD-L1 upregulation at the transcriptional level in cancer cells.

**Figure 3 cancers-13-04426-f003:**
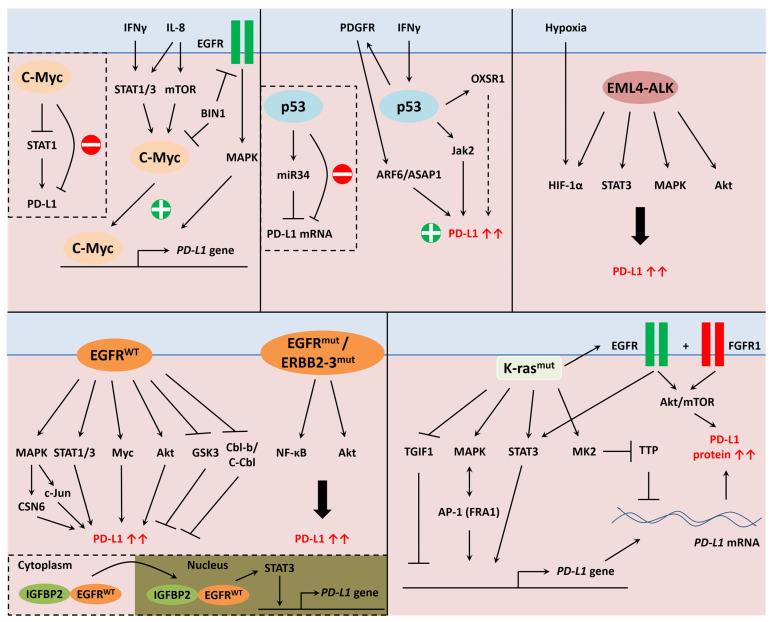
Oncogenic regulation of PD-L1 in cancer cells. Oncogenes including c-Myc, p53, EML4-ALK fusion gene, EGFR, and K-ras regulate PD-L1 expression, positively or negatively, by various and distinct mechanisms.

**Figure 4 cancers-13-04426-f004:**
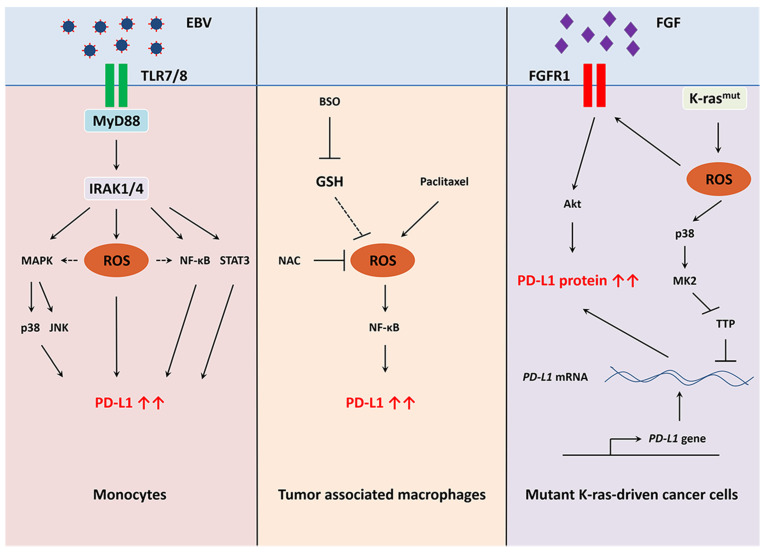
Redox regulation of PD-L1 in EBV-infected monocytes, tumor-associated macrophages, and mutant K-ras-driven cancer cells. ROS induce PD-L1 expression by different mechanisms in immune and cancer cells. EBV infection activates MyD88/IRAK/ROS signaling and induces PD-L1 expression in monocytes. Paclitaxel induces ROS-mediated PD-L1 upregulation in tumor associated macrophages. Mutant K-ras causes an increase in ROS production in cancer cells. This oxidative stress stimulates FGFR1 signaling and PD-L1 mRNA stability.

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
