# Peer review of "The Role of Oncogenes and Redox Signaling in the Regulation of PD-L1 in Cancer"

_cancers, 2021, doi:10.3390/cancers13174426_

Round 1

Reviewer 1 Report

Thank you for allowing me to read this article. From my point of view, it is well written. Even though there is a lot of information, from my point of view it is well structured. Although this is not the first summary article on this topic, the authors present new perspectives as well as new information. The article also fulfills the aims. In the article, I see the potential for readers and research teams to use for further research. What I miss a bit as a clinician is more space for discussion about usability in clinical practice. On the other side, I understand that this was not the main subject of the review article, and the authors touched on the issue a little bit. I also appreciate the figures which make it easier to follow such a complex issue of the article. For the above reasons, I recommend accepting the article. 

Author Response

Thank you for allowing me to read this article. From my point of view, it is well written. Even though there is a lot of information, from my point of view it is well structured. Although this is not the first summary article on this topic, the authors present new perspectives as well as new information. The article also fulfills the aims. In the article, I see the potential for readers and research teams to use for further research. What I miss a bit as a clinician is more space for discussion about usability in clinical practice. On the other side, I understand that this was not the main subject of the review article, and the authors touched on the issue a little bit. I also appreciate the figures which make it easier to follow such a complex issue of the article. For the above reasons, I recommend accepting the article.

Response: We would like to thank the reviewer for his/her positive comments, and agree that it would be of high interests to further discuss the clinical implications of PD-L1 regulation. Although the main idea of this review article is to provide an up-to-date overview regarding the regulation of PD-L1 expression in term of mechanisms, this topic would also interest clinicians since PD-L1 expression in cancer cells is an important factor that induces immune evasion and potentially influences clinical response to therapy with immune checkpoint antibodies. In the context of clinical relevance, the up-regulation of PD-L1 by ROS/redox signaling and oncogenes such as K-ras is of particular significance, since ROS stress and activation of K-ras are frequently observed in various types of human cancers. Furthermore, because ROS play a major role in up-regulation of PD-L1 expression, modulation of redox signaling could be a potential strategy to alter tumor immune microenvironment and improve the efficacy of immunotherapy. Such therapeutic strategy is novel and requires further evaluation in clinically-relevant settings. In the “conclusions” section (page 16), we have added some discussion to make these points.

Reviewer 2 Report

Summary

In this review Glorieux et al survey various aspects of PD-L1 regulation and activity in cancer. The manuscript starts from a general introduction to the key discoveries that led to the development of immune checkpoint blockade therapeutics, describing the seminal work of James Allison and Tasuku Honjo. The authors then continue to describe the structure, evolution, cellular and tissue distribution of PD-L1, as well as the biological function and related diseases linked to PD-L1. The second part of the paper is focuses on PD-L1 regulation across different regulatory modalities, with emphasis on specific oncogenes.

The review provides a detailed view of PD-L1 role and regulation. It is well written, though at times reads as a list of facts where it is recommended to highlight a narrative to orient the reader. There are some statements that need to be mitigated or altered to reflect the latest findings and acknowledge the complexity and open questions concerning PD-L1 biology. It is critical to make a distinction between association and causation, particularly when describing interventions that increase PD-L1 expression as means to sensitize tumors to ICB. These effects are highly unlikely to be mediated by PD-L1 itself, and in this context, it is likely to merely be a surrogate marker of T cell infiltration and/or cell autonomous immune responses. Below are more specific comments along these lines.

Major comments

1) The prognostic value of PD-L1 expression in predicting clinical response to ICB has been debatable. In numerous studies PD-L1 has been found to have no significant predictive value in predicting ICB responses (e.g., Brahmer et al., N Engl J Med 2015; Hanna et al., JCI Insight. 2018; Jerby-Arnon et al., Cell 2018; Carbone et al., N Engl J Med. 2017). The authors should state that and clarify that predicting ICB responses is still a major challenge in immunoncology.

2) As for general survival, high expression of PD-L1 is often associated with improved survival (KM derived using https://kmplot.com/analysis/index.php?p=service are attached as evidence). As PD-L1 is activated in immune-rich and cytokine-rich environments, it is likely that further stratifications of the patient populations are needed to deconvolve these multifactorial effects on survival. It is highly recommended that the authors avoid making unsound claims concerning PD-L1 value as a biomarker, both in the ICB and general setting.

3) As alluded to by the authors, PD-L1 is tightly co-regulated with immune responses as IFNg, the JAK-STAT pathway, and antigen presentation. PD-L1 function as a negative regulator of immune response and inflammation occurs in conjunction to the cellular sensing and response to these external cues. Thus, PD-L1 is also associated with “hot” TME and immune-responding cancer cells. Moreover, PD-L1 is likely to function as a secondary resistance mechanism, in immune-rich tumors. Immune-deprived tumors usually do not exhibit high PD-L1 expression, as there is not any selective advantage in expressing it. However, it is those tumors that are substantially more challenging to treat. It is recommended to surface these aspects and distinguish between causation and association in this context.

4) At the conclusion section the authors raise the possibility that up-regulating PD-L1 can be a path to sensitize tumors to ICB. This is equivalent to claiming that introducing BRAF mutations will make melanoma tumors sensitive to BRAF inhibitors. Compounds that up-regulated PD-L1 can indeed in some cases sensitize the tumor to ICB, but this is unlikely to be in a PD-L1-mdiated manner. Instead, it is likely to be the result of triggering other processes which are co-regulated with PD-L1 (e.g., IFN response and antigen presentation).

5) As alluded to by the authors, it is still unclear whether p53 and c-Myc are positive or negative regulators of PD-L1, and it might be that their function is context-dependent. To orient the reader, it is recommended to state that at the beginning of the sections “5.7.4. Protein p53” and “5.7.3. c-Myc”.

Minor comments:

1) The sentences in rows 210-212 are somewhat unclear: “It is interesting to note that mutations exist in the coding sequence. Only the mutation G146A was reported in hepatocarcinoma.”

2) Row 306: “stabilize PD-L1 mRNA stability”, should remove the work “stability.

Author Response

Summary

In this review Glorieux et al survey various aspects of PD-L1 regulation and activity in cancer. The manuscript starts from a general introduction to the key discoveries that led to the development of immune checkpoint blockade therapeutics, describing the seminal work of James Allison and Tasuku Honjo. The authors then continue to describe the structure, evolution, cellular and tissue distribution of PD-L1, as well as the biological function and related diseases linked to PD-L1. The second part of the paper is focuses on PD-L1 regulation across different regulatory modalities, with emphasis on specific oncogenes.

The review provides a detailed view of PD-L1 role and regulation. It is well written, though at times reads as a list of facts where it is recommended to highlight a narrative to orient the reader. There are some statements that need to be mitigated or altered to reflect the latest findings and acknowledge the complexity and open questions concerning PD-L1 biology. It is critical to make a distinction between association and causation, particularly when describing interventions that increase PD-L1 expression as means to sensitize tumors to ICB. These effects are highly unlikely to be mediated by PD-L1 itself, and in this context, it is likely to merely be a surrogate marker of T cell infiltration and/or cell autonomous immune responses. Below are more specific comments along these lines.

Response: We would like to thank the referee for the constructive comments and suggestions, which are highly valuable and helpful for us to strengthen our manuscript. The important points regarding making a distinction between association and causation and the role of PD-L1 in immunotherapy are discussed according to the reviewer’s suggestions as described in detail below.

Major comments

1) “The prognostic value of PD-L1 expression in predicting clinical response to ICB has been debatable. In numerous studies PD-L1 has been found to have no significant predictive value in predicting ICB responses (e.g., Brahmer et al., N Engl J Med 2015; Hanna et al., JCI Insight. 2018; Jerby-Arnon et al., Cell 2018; Carbone et al., N Engl J Med. 2017). The authors should state that and clarify that predicting ICB responses is still a major challenge in immunoncology.

Response: We agree with the reviewer, and have deleted the statement on prognostic value of PD-L1 expression in the summary, abstract and main text. Moreover, we have now briefly discussed on page 3 that PD-L1 as a biomarker of predicting immunotherapy responses is still a subject of debate, and we added the above suggested references.

2) “As for general survival, high expression of PD-L1 is often associated with improved survival (KM derived using https://kmplot.com/analysis/index.php?p=service are attached as evidence). As PD-L1 is activated in immune-rich and cytokine-rich environments, it is likely that further stratifications of the patient populations are needed to deconvolve these multifactorial effects on survival. It is highly recommended that the authors avoid making unsound claims concerning PD-L1 value as a biomarker, both in the ICB and general setting.

Response: Thank you for the comments concerning PD-L1 value as a biomarker. The cited databases predicting patient survival mainly based on mRNA levels. In this review, we pointed out that PD-L1 mRNA often does not correlate with PD-L1 protein levels. For instance, a tumor cell might have relatively low PD-L1 mRNA but higher PD-L1 protein due to its stability. Thus, the results from such databases are difficult to interpret, as we described after the section “5.7. Oncogenes”. We have modified the relevant sentences in the manuscript to avoid potential confusion regarding PD-L1 value as a biomarker of ICB responses.

3) “As alluded to by the authors, PD-L1 is tightly co-regulated with immune responses as IFNg, the JAK-STAT pathway, and antigen presentation. PD-L1 function as a negative regulator of immune response and inflammation occurs in conjunction to the cellular sensing and response to these external cues. Thus, PD-L1 is also associated with “hot” TME and immune-responding cancer cells. Moreover, PD-L1 is likely to function as a secondary resistance mechanism, in immune-rich tumors. Immune-deprived tumors usually do not exhibit high PD-L1 expression, as there is not any selective advantage in expressing it. However, it is those tumors that are substantially more challenging to treat. It is recommended to surface these aspects and distinguish between causation and association in this context.

Response: We thank the reviewer for these insightful comments. As suggested, these points are now discussed in revised manuscript (page 4), and a relevant reference has been added (PMID: 33859752).

4) “At the conclusion section the authors raise the possibility that up-regulating PD-L1 can be a path to sensitize tumors to ICB. This is equivalent to claiming that introducing BRAF mutations will make melanoma tumors sensitive to BRAF inhibitors. Compounds that up-regulated PD-L1 can indeed in some cases sensitize the tumor to ICB, but this is unlikely to be in a PD-L1-mediated manner. Instead, it is likely to be the result of triggering other processes which are co-regulated with PD-L1 (e.g., IFN response and antigen presentation).

Response: We agree with the reviewer that it is not a good idea to artificially up-regulate PD-L1 expression in cancer cells in order to increase their sensitivity to ICB. In the conclusion section we did not mean to raise the possibility to up-regulate PD-L1 expression in cancer cells to sensitize tumors to immunotherapy. In some preclinical studies cited in this review, the authors discovered new mechanisms of PD-L1 regulation and tried to overexpress PD-L1 to sensitize tumors to antibodies. Our point was that while these studies are important with respect to mechanisms of PD-L1 regulation and potential clinical implications, it is not a good therapeutic strategy to artificially induce PD-L1 expression in cancer. That is the reason we wrote in the next sentence “this strategy appears to be a double-edged sword as it could potentially enhance tumor immunosuppression.” It is important to make a clear distinction between the intrinsic expression levels of PD-L1 in the tumor tissues and artificial induction of PD-L1 expression. Tumor tissues with high intrinsic expression of PD-L1 tend to be immunosuppressive, which could be overcome by therapeutic antibodies, whereas artificial induction of PD-L1 expression would promote immunosuppression, which should be avoided. We have revised the text to avoid misunderstanding (pages 15-16).

5) As alluded to by the authors, it is still unclear whether p53 and c-Myc are positive or negative regulators of PD-L1, and it might be that their function is context-dependent. To orient the reader, it is recommended to state that at the beginning of the sections “5.7.4. Protein p53” and “5.7.3. c-Myc”.

Response: As suggested, we have moved these sentences at the beginning of their section (pages 12-13).

Minor comments

1) The sentences in rows 210-212 are somewhat unclear: “It is interesting to note that mutations exist in the coding sequence. Only the mutation G146A was reported in hepatocarcinoma.”

Response: Indeed, this sentence was unclear and has now been deleted (page 6).

2) Row 306: “stabilize PD-L1 mRNA stability”, should remove the work “stability.

Response: The word “stability” is redundant and has been removed (page 8).

Reviewer 3 Report

This comprehensive review describes the regulation of expression of PD-L1 in cancer by intrinsic factors. Even though the details provided are many and accurate, the clinical perspective is lacking. In particular, we know that the presence of activated oncogenes reduces the clinical response to immunological checkpoint inhibitors but this is not commented on. Consequently, the usefulness of this review for the clinical reader is unclear.

Author Response

“This comprehensive review describes the regulation of expression of PD-L1 in cancer by intrinsic factors. Even though the details provided are many and accurate, the clinical perspective is lacking. In particular, we know that the presence of activated oncogenes reduces the clinical response to immunological checkpoint inhibitors but this is not commented on. Consequently, the usefulness of this review for the clinical reader is unclear.”

Response: Thank you for making the important point regarding the need to discuss the clinical perspective. Although the main purpose of this review article is to provide an up-to-date overview regarding the regulation of PD-L1 expression in term of mechanisms, it would be important to discuss the clinical relevance of these mechanisms and potential therapeutic implications. Since PD-L1 expression in cancer cells is an important factor that potentially influences clinical response to therapy with immune checkpoint antibodies, it is plausible to hypothesize that molecules such as oncogenes that regulate PD-L1 expression would affect the clinical response to immune checkpoint inhibitors. It seems that the up-regulation of PD-L1 by ROS/redox signaling and oncogenes is of particular significance, since ROS stress and activation of oncogenes are often observed in various types of human cancers. Furthermore, because ROS play a major role in up-regulation of PD-L1 expression, modulation of redox signaling could be a potential strategy to alter tumor immune microenvironment and improve the efficacy of immunotherapy. Such therapeutic strategy is novel and requires further evaluation in clinically-relevant settings. In the “conclusions” section (page 16), we have added some discussion to make these points.

Round 2

Reviewer 3 Report

The Authors have attempted to provide an answer to the questions by introducing a short note in the conclusion. However, this insertion is insufficient to clarify the problem and I suggest to integrate the text, for each oncogene described in the article, with a statement with an accompanying bibliography, which describes the possibilities of administering an immunotherapy in that specific mutational context.

Author Response

The Authors have attempted to provide an answer to the questions by introducing a short note in the conclusion. However, this insertion is insufficient to clarify the problem and I suggest to integrate the text, for each oncogene described in the article, with a statement with an accompanying bibliography, which describes the possibilities of administering an immunotherapy in that specific mutational context.

Response: We appreciate the reviewer’ comment regarding the exact role or impact of each individual oncogene on tumor immune response. The limited information currently available in this area is insufficient as a basis to formulate optimal immunotherapeutic strategies. Furthermore, since some tumor may have mutations in multiple genes such as K-ras and p53 in the same cells, this would make it even more complicated to formulate oncogene/mutation-specific immunotherapy. Obviously, further studies are needed in this area. In the section where individual oncogenes are described, we now added comments on their impact on immune response and potential therapeutic strategies according to available information. The modifications have been highlighted in blue color on pages 9, 12-14, and 17.

Round 3

Reviewer 3 Report

The Authors have attempted to reply to the Reviewer's requests by adding comments and clarifications to the manuscript. Although the translational relevance of the manuscript is improved slightly, it can be accepted in the present form.